



# Characterization of the unsteady aerodynamic response of a floating offshore wind turbine

Simone Mancini[1], Koen Boorsma[2], Marco Caboni[2], Marion Cormier[3], Thorsten Lutz[3], Paolo Schito[1], and Alberto Zasso[1]

[1]Politecnico di Milano, Department of Mechanical Engineering, Milano, Italy
[2]ECN part of TNO, Petten, Netherlands
[3]University of Stuttgart, Institute of Aerodynamics and Gas Dynamics, Stuttgart, Germany

**Correspondence:** Simone Mancini (simone1.mancini@mail.polimi.it)

**Abstract.** The disruptive potential of floating wind turbines has attracted the interest of both industry and scientific community. Lacking a rigid foundation, such machines are subject to large displacements whose impact on the aerodynamic performance is not yet fully acknowledged. In this work, the unsteady aerodynamic response to an harmonic surge motion of a scaled version of the DTU10MW turbine is investigated in detail. The imposed displacements have been chosen representative of typical

platform motions. The results of different numerical models are validated against high fidelity wind tunnel tests specifically focused on the aerodynamics. Also a linear analytical model, relying on the quasi-steady assumption, is presented as a theoretical reference. The unsteady responses are shown to be dominated by the first surge harmonic and a frequency domain characterization, mostly focused on the thrust oscillation, is conducted involving aerodynamic damping and mass parameters. A very good agreement among codes, experiments and quasi-steady theory has been found clarifying some literature doubts. A

convenient way to describe the unsteady results in non-dimensional form is proposed, hopefully serving as reference for future work.

## 1 Introduction

Lacking a rigid foundation, floating offshore wind turbines (FOWTs) are subjected to large displacements during their operation. Therefore, the classical control strategies, suitable for bottom-fixed turbines, have to be redesigned accounting for

these motions. The application of an inland turbine controller to a FOWT might lead, indeed, to dangerous controller induced resonances (Nielsen et al., 2006). Moreover, Sebastian and Lackner (2013) pointed out that floater displacements can be a major source of aerodynamic unsteadiness, because their typical periods are comparable to the time scale of dynamic inflow ($\tau = D/V_0$). Since the design of a FOWT controller cannot prescind from an accurate inflow modelling (Pedersen, 2017), the presence of dynamic inflow effects due to platform motions requires a detailed investigation.

Depending on the type of floater, different degrees of freedom (D.o.F.) are more excited than others. For example, semi-submersible (e.g. WindFLoat®) and tension-leg platforms (TLPs) are less affected by pitch oscillations than spar type floaters (e.g. HYWIND™). However, owing to the typical alignment between wind and waves, platform surge is commonly one of the most significant modes. In addition, small turbine pitch rotations are often approximated to surge motions by means of





linearization. Despite the simple kinematics consents the use of momentum theory with little modifications, it is still unclear

whether current blade element momentum (BEM) codes can adequately model FOWT aerodynamic response to surge. In fact, neither the impact of unsteady effects nor the accuracy of current engineering dynamic inflow models are uniquely acknowledged for this case.

Several numerical studies addressing the impact of surge motion on turbine performances have been conducted. Regardless the common benchmark provided by the NREL's 5MW reference wind turbine (RWT), the results led to rather discordant

conclusions. Studying characteristic floater motions with a free vortex wake (FVW) code, Sebastian and Lackner (2012) underlined the need of higher fidelity models than BEM. Conversely, de Vaal et al. (2014) found that surge displacements, in the typical frequency range of a TLP, were slow enough for dynamic inflow effects to be insignificant. Such conclusion was drawn comparing a moving actuator disk (AD) to both a quasi-steady BEM and another BEM with Øye's dynamic inflow model (1990) implemented. At a similar frequency though, Micallef and Sant (2015) found relevant differences among BEM,

generalized dynamic wake (GDW) and AD model results. They also noticed that unsteadiness increased with the tip speed ratio ($\lambda$). This was confirmed by a FVW code too (Farrugia et al., 2016). The most detailed work on the aerodynamic effects of surge was performed by Tran and Kim (2016), who were the first to adopt a full CFD model for the purpose. Considering similar surge cases to de Vaal et al., they solved the RANS equations with a $k - \omega$SST model featuring an overset mesh technique; the results were then compared against a BEM with Øye's model and a GDW solver. The discrepancies at the highest

frequencies and amplitudes aroused doubts on possible dynamic inflow effects. Unfortunately, none of these studies addressed closely the theme of control, limiting the analysis to the time domain. From the controller design point of view though, the characterization of load signals in terms of amplitudes and phases in frequency domain is fundamental because it allows to evaluate control-relevant parameters like the aerodynamic damping ($c_{aero}$), which rules system dynamics in surge. One of the few works linking FOWT control and unsteady aerodynamics due to platform motions was that of Lennie et al. (2016), but the

aim was other than spotting the influence of dynamic inflow. In fact, it studied the effects of a feathering blade pitch actuation during turbine pitch motion (approximated to surge) by means of a FVW model.

Also the lack of experimental data for code validation hampered a clear understanding of dynamic inflow effects due to surge. Most of the available works involved Froude scaled models, tested in water basins equipped with fans to reproduce the wind. Apart from some tests on very small turbines (Farrugia et al., 2014; Sant et al., 2015; Khosravi et al., 2015), from which

it was hard to draw any full-scale conclusion, a validation campaign on a 1:60 scaled version of the NREL5MW RWT was conducted by Ren et al. (2014), but the interest was mainly on the hydrodynamic loading and surge was considered as an output. At MARIN's offshore basin Goupee et al. (2017) carried out intensive testing of a 1:50 model of the NREL5MW, mounted on a semi-submersible platform, specifically focusing on control aspects. More recently, Madsen et al. (2020) performed similar experiments on a 1:60 model of the DTU10MW RWT mounted on a TLP, investigating the system's response to various

wind and waves conditions with different control strategies. Nevertheless, it is very difficult to understand the influence of unsteady aerodynamics from combined wind and waves tests. For this reason, Polimi decided to focus more specifically on the aerodynamics, aiming both at an increased comprehension and at the generation of valuable data for codes validation. For the purpose, a 1:75 model of the DTU10MW RWT was designed within LIFES50+ project (Bayati et al., 2017a, c). The scaled





turbine was mounted inside Polimi's wind tunnel (GVPM) on a two D.o.F. test rig allowing to impose both pitch and surge
motions. The first experiments conducted seemed to show relevant traces of unsteady effects due to surge (Bayati et al., 2016).
However, after a thorough revision, it was understood that the results had been strongly biased by tower flexibility. Therefore,
a stiffer tower was manufactured to run new surge tests in project UNAFLOW.

UNAFLOW (UNsteady Aerodynamics for FLOating Wind) was a collaborative project, belonging to the EU-IRPWIND
program, that involved four research institutions: POLIMI, ECN (now part of TNO), USTUTT and DTU. It focused on ad-
vanced aerodynamic modelling and novel experimental approaches for studying the unsteady behaviour of multi-megawatt
floating turbine rotors (2018). The work, carried out between June 2017 and April 2018, was divided in two work packages:
the first studied the 2-dimensional airfoil aerodynamics from DTU Red wind tunnel tests; the second focused on the scaled
model performances under imposed surge motion, comparing GVPM experiments with numerical simulations. The numerical
part involved a full CFD model, provided by USTUTT, plus a BEM and a free vortex code (AWSM) provided by TNO. Input to
the lifting line codes were the airfoil polars obtained in the first work package. The significant amount of data generated within
UNAFLOW was made available to the scientific community, including a number of steady and unsteady tests on SD7032
airfoil, steady and unsteady full turbine loads and PIV wake measurements. Latter were investigated by (Bayati et al., 2017b,
2018b) and an overview of the main results was given in Bayati et al. (2018a). Concerning CFD results, only those obtained
with the axisymmetric model were published in Cormier et al. (2018) and included in the final project report (2018). More-
over, an inconsistency in the set up of several simulations was later discovered, explaining the large discrepancies found in the
comparison. For this reason, a complete results revision and update has been recently conducted to reach a final convergence
(Mancini, 2020).

In this work the latest comparison of the turbine performances during surge is presented. Diversely from the original UN-
AFLOW report (2018): the unsteady thrust response from wind tunnel measurements has been obtained with a revised inertia
subtraction procedure; the full CFD results have been included, together with new BEM and AWSM simulations; the out-
comes of an Actuator Line code (AL) have been added as an intermediate fidelity level. A frequency domain analysis has
been performed focusing on control-relevant quantities and the influence of surge motion's amplitude and frequency has been
investigated. To have a theoretical reference, a simple linear model based on quasi-steady theory (Appendix A) has also been
included in the comparison. In attempt of giving a more general representation to the unsteady analysis, the results in frequency
domain have been reported in non-dimensional form, defining some meaningful parameters that may be used conveniently in
future work. This paper aims to shed light on the surge induced unsteady aerodynamics of a FOWT, providing the first publicly
available experimental data to be used as a benchmark for codes validation. The main research goal was to reach a clearer
awareness on the impact of dynamic inflow effects. As side benefits, also a valuable comparison among the different fidelity
models for wind turbine aerodynamic modelling along with a robust result nondimensionalization strategy have been produced.





**Table 1.** Key parameters of the DTU 10MW compared to Polimi's model.

|  | **DTU10MW RWT** | **Polimi model** |
| --- | --- | --- |
| **Control** | Variable speed + Collective pitch | Variable speed + individual pitch |
| **Drivetrain** | Medium Speed, multiple stage gearbox | Transmission belt, epicyclic gearbox |
| **Gearbox ratio** | 50 | 42 |
| **Diameter** | 178.3 m | 2.38 m |
| **Hub height** | 119 m | 2.05 m |
| **Tilt angle** | 5 ° | 5 ° |
| **Coning angle** | -2.5 ° | 0 |
| **Blade prebend** | 3.33 m | 0 |

## 2   Wind tunnel tests

The turbine model tested in UNAFLOW was a 1:75 scaled version (2.38 m diameter) of the DTU10MW RWT. Such reference rotor was chosen to resemble the size of current offshore units being installed. The model was completely designed and engineered by Politecnico di Milano within LIFES50+, pursuing an accurate match of the RWT aerodynamic coefficients, thrust coefficient ($C_T$) especially, because of the leading role of thrust in the dynamics of a FOWT. Whilst in Froude scaled models (e.g. Goupee et al., 2017; Madsen et al., 2020) in order to cope with the steady thrust reduction due to lower Reynolds the blade pitch is typically adjusted, here a different approach was followed for a better aerodynamic accuracy. Provided that the dimensions were scaled by a factor 75 to fit in the wind tunnel and the wind velocity was scaled by a factor 3 for surge actuation purposes, the Reynolds number was 225 times lower than reality. Hence, a low Re profile (SD7032) was employed changing chord and twist distributions to fulfill the loads compliance. Such procedure allowed to achieve an accurate thrust reproduction throughout the whole operating range, together with a satisfactory torque match up to rated conditions (Bayati et al., 2017c). The scaled turbine was also equipped with variable speed and individual blade pitch controllers (Bayati et al., 2017a), but these features were not exploited in the unsteady tests. In Table 1 the key characteristics of the scaled turbine are compared to those of the RWT.

The experimental campaign was carried out in the Boundary Layer Test Section of the GVPM (13.84 m wide x 3.84 m high x 35 m long). The tests were performed in empty inlet configuration (i.e. without roughness elements or turbulence generators), aiming to obtain an inflow velocity profile as constant as possible. Figure 1 shows the resulting wind speed and turbulence intensity (T.I.) distributions measured 5 m upstream the rotor plane and normalized by the value measured at hub height. The wind speed could be considered constant in the rotor zone with a T.I. around 2 %.





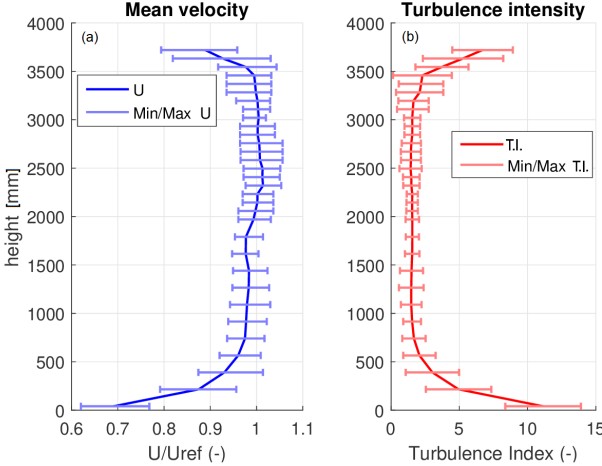

**Figure 1.** Wind tunnel speed and T.I. profiles normalized by the value measured at hub height with error bars.

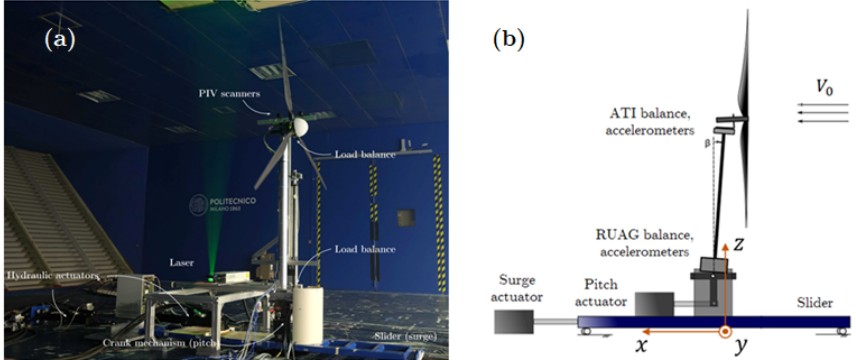

**Figure 2.** Experimental setup picture (a) and schematic sketch with reference system (b).

## 2.1 Experimental set-up

110   The model turbine was mounted on a slider, commanded by means of an hydraulic actuator to produce the desired surge motion, as shown in Fig. 2; a schematic sketch with the reference system adopted in this work is also included. Another hydraulic piston was connected to a slider-crank mechanism underneath the tower, which allowed to control turbine pitch too. However, latter feature was not exploited in UNAFLOW and the mechanism was only used to place the rotor perpendicular to the inflow, i.e. to cancel the 5 ° design tilt angle. This choice was made to avoid the periodic effects related.

115   A wide array of sensors was employed to measure both the dynamic response and the flow field characteristics. All the instruments were synchronized and everyone sampled at 2 KHz. The shaft was equipped with an encoder and a proximitor, measuring the rotational speed and azimuthal position respectively. Loads were measured by means of two six-components balances mounted one at the tower base (RUAG) and one underneath the nacelle (ATI); only the latter was used in the post-



**Table 2.** Operating conditions tested in UNAFLOW.

|  | $V_0 \, [m \, s^{-1}]$ | $\Omega \, [\mathrm{rpm}]$ | $\lambda \, [-]$ | $\theta_p \, [°]$ |
|---|---|---|---|---|
| **RATED1** | 2.5 | 150 | 7.5 | 0 |
| **RATED2** | 4.0 | 241 | 7.5 | 0 |
| **ABOVE** | 6.0 | 265 | 5.5 | 12.5 |

processing. A couple of accelerometers was placed next to each balance: at the base they measured along surge and heave directions (x and z according to Fig. 2b); at the nacelle along surge and sway (x and y). To measure base's surge position, both an LVDT and a laser transducer were placed. The laser was chosen as reference measure for its lower delay. For what concerns the flow field, the incoming wind speed was measured by a Pitot tube located 5m upstream the turbine at a height of 1.5 m from the floor. The PIV system consisted of a pair of cameras mounted on an automatic traversing system and connected to an Nd:Yag laser, which enlightened the seed particles in the flow. The pictures were post-processed with PIVview 3C (PIVTEC) to create the 2D velocity contours in various zones of the near wake. This work focuses on the aerodynamic loads and the wake measurements, despite tightly linked, will not be considered.

## 2.2 Steady tests

Before imposing the surge motion, steady tests were carried out at three different wind speeds obtaining the reference scaled model's stationary performances. The operating conditions considered are reported in Table 2. The first two cases (named RATED1 and RATED2) were both at optimal tip speed ratio ($\lambda = 7.5$), with the blades in neutral pitch position but with different wind velocities (variable speed rotor). The ABOVE case instead, considered an above-rated wind speed with lower $\lambda$ and a blade pitch of 12.5 ° towards feather.

## 2.3 Unsteady tests

For all the three steady conditions a number of unsteady tests was performed. An hydraulic actuator was used to impose a sinusoidal surge motion to the slider upon which the turbine was mounted. The displacement at the base of the tower could be expressed as:

$$x_B(t) = A_s \sin 2\pi f_s t; \tag{1}$$

being $A_s$ and $f_s$ the surge amplitude and frequency respectively. Different pairs of amplitude and frequency values were tested. Being the platform surge of a FOWT induced by the hydrodynamics, the frequency range of motion depends on the waves excitation. Therefore, different $f_s$ were chosen to represent possible frequencies at which a peak in the sea waves' spectrum might be found. The selected range went from 0.125 Hz to 2 Hz in model scale, corresponding to $0.005 \leq f_s^{real} \leq 0.08$ Hz in full scale. The range was totally consistent to those investigated in literature (de Vaal et al., Micallef and Sant, Farrugia et al. and Tran and Kim). Provided that real turbine oscillation amplitudes depend on the floater type and on site specific





parameters (e.g. water depth and mooring lines), different $A_s$ were considered at each frequency so as to cover a wide range
of possibilities. The amplitude range selected guaranteed the magnitude of the angle of attack variation in surge to be limited,
confining dynamic stall effects to the blade root only. A total of 84 unsteady tests was conducted, 28 for each steady operating
condition. The full test matrix can be found in Bernini et al. (2018). It is important to observe that, during surge, the standard
turbine controller was not active and both the blade pitch and the rotational speed were kept constant at the values reported in
Table 2.

One of the major challenges of the experimental campaign was the extraction of the aerodynamic thrust from the balance
measurements. In fact, especially at the higher $f_s$, the load signal was heavily affected by nacelle inertial contribution caused by
the imposed surge acceleration. Originally the inertia subtraction was made assuming a perfectly rigid system: the aerodynamic
part was extracted subtracting from the force signal measured during surge, the force measurement obtained with the same surge
motion but without wind. Tests without wind were referred to as NOW (i.e. NO-Wind). Mancini (2020) showed that the high
aerodynamic damping, generated by the rotor when the wind was active, had led to dynamic amplification effects that biased
LIFES50+ results. The stiffer tower employed in UNAFLOW was proven able to mitigate such effects. However, an alternative
inertia subtraction procedure capable of reducing the bias due to flexibility was proposed and it has been used in this work.
Having the acceleration measure in $x$ direction (Fig. 2b) at the nacelle, the aerodynamic thrust force has been obtained as:

$$T(t) = F^{ATI}(t) + m\, ACC(t)\, ; \tag{2}$$

being $T$ the aerodynamic thrust, whilst $F^{ATI}$ and $ACC$ respectively the ATI balance's and the accelerometer's measurements
along $x$. The mass of the nacelle ($m$), i.e. all what attached to the ATI balance, has been estimated from the NO-Wind tests
considering the amplitudes of the surge frequency harmonics extracted through a Fast Fourier Transform:

$$m = \frac{|F^{ATI}_{NOW}|_{@f_s}}{|ACC_{NOW}|_{@f_s}}\, . \tag{3}$$

A comparison among different inertia subtraction procedures can be found in Mancini (2020).

In order to avoid leakage in the frequency domain analysis all wind tunnel test signals have been windowed, always consid-
ering six full surge periods.

## 3  Numerical codes description

Four different numerical methods have been used for numerical-experimental cross validation: a BEM and a FVW (AWSM)
part of ECN's Aero Module (Sect. 3.1), an AL (Sect. 3.2) and a full CFD model (Sect. 3.3). The codes have been selected to
cover almost the whole state-of-art fidelity range available for wind turbine aerodynamic modelling. This way it is possible to
better understand the capability of each method to deal with unsteady aerodynamics.

### 3.1  Aero Module

The ECN Aero-Module (Boorsma et al., 2011, 2016, 2020) contains two aerodynamic models, namely the Blade Element
Momentum (BEM) method similar to the implementation in PHATAS (Lindenburg and Schepers, 2000) and a free vortex





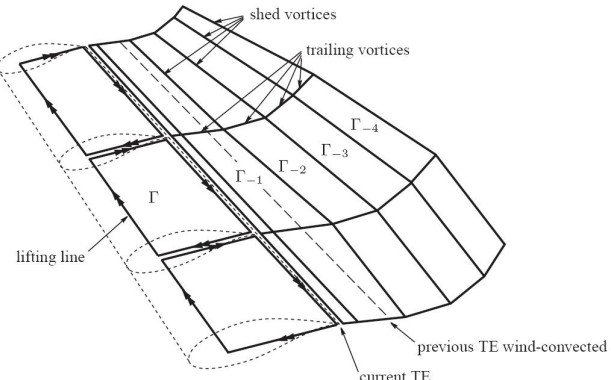

**Figure 3.** AWSM wake geometry (van Garrel, 2003).

wake code in the form of AWSM (van Garrel, 2003). Both models are lifting line codes, i.e. they make use of aerodynamic look-up tables to evaluate airfoil performance. Several dynamic stall models, 3D correction models, wind modeling options and a module for calculating tower effects are included. The set-up allows to easily switch between the two aerodynamic models whilst keeping the external input the same, which is a prerequisite for a good comparison between them.

### 3.1.1 Wake modeling

Since a pure BEM code only resolves the rotor plane, an engineering model has to be added to simulate wake effects. Therefore the ECN dynamic inflow model (Snel and Schepers, 1994) has been implemented to account for the aerodynamic rotor 'inertia'. The dynamic inflow model adds another term to the axial momentum equation. This term is proportional to the time derivative of the annulus averaged axial induction of the element under consideration and its magnitude varies with the radial position.

AWSM uses the blade geometry to create vortex lattices which are convected in the wake, conserving shed and trailing vorticity as depicted in Fig. 3. Here the trailing vorticity accounts for the effects of spanwise circulation variation, whilst the shed vorticity accounts for the effects of bound vortex variation with time. Consequently all wake related flow phenomena (e.g. dynamic inflow, aero-elastic instabilities featuring shed vorticity variation and skewed wake effects) are modeled intrinsically, where they are covered by engineering models or not covered at all in BEM. If the wake points are modeled free, the convection of each wake point is determined by the aggregate of the induced velocities from all vortices using the Biot and Savart law.

### 3.1.2 BEM implementation in surge

A turbine subjected to surge or pitching motion experiences apparent wind velocities at the rotor due to the movements of the tower base. Since these wind velocities add energy to the system (as they are induced by the waves) it can be argued to incorporate these in the effective wind speed used in the momentum part of the BEM equations. This in addition to the obvious implementation of this relative motion in the element part of the BEM equations. The validity of this statement is verified by comparing a simulation in surge with a moving rotor (which is used in the present work) to a simulation with a 'fixed' rotor



featuring a (sinusoidal) wind variation in agreement with the surge motion. Free vortex wake simulations give nearly identical results for both approaches in this case, indicating that the main effect the wind turbine rotor experiences is the apparent wind effect rather than the rotor moving into and out of its own wake. For the BEM simulations it is observed that the shape of the force response is off if apparent wind velocities are not taken into account in the momentum equations.

Implementation wise this can result in a challenge since an aero-elastic code is not always aware whether motion of the blade is due to flexible nature of the turbine (e.g. tower fore-aft bending) or due to platform motion induced by waves. Recommended practice here is to register translational and rotational movement at the tower base and extrapolate the resulting apparent wind velocities to the designated rotor plane locations. For a pitching movement this would imply a linear variation with height of apparent wind velocity over the rotor disk, and hence a non-uniform inflow condition which anyhow is a challenge for BEM
simulations.

### 3.1.3   Aero Module settings

A rigid version of the turbine has been simulated. The airfoil data have been obtained from the corresponding 2D experiment in UNAFLOW (Bernini et al., 2018) for clean conditions at a Reynolds number of $1 \cdot 10^5$. The time step has been kept at the approximate equivalent of 10 ° azimuth for both the BEM and AWSM simulations. The Snel dynamic stall model (Snel, 1997)
has been applied to all simulations.

For the free vortex wake simulation, the number of wake points has been chosen to make sure that the wake length developed over at least three rotor diameters downstream of the rotor plane. The wake convection has been set free for approximately two rotor diameters downstream. For the remaining diameter in the far wake, the blade average induction at the free to fixed wake transition is applied to all wake points.

## 3.2   Actuator line

The actuator line model has been chosen as an intermediate step between free vortex method and full CFD. To run the simulations MECC's (Polimi) in-house developed actuator line code for OpenFOAM (2011) has been used. Diversely from classical actuator line, such code adopts an effective velocity model (EVM), proposed by Schito and Zasso (2014), to evaluate the relative velocity vector used in the calculation of the aerodynamic forces. In particular, instead of evaluating it at the very same
point where the force is applied, the EVM considers a series of sampling points along a line, placed perpendicularly to the wind and upstream of the profile leading edge, and estimates the relative velocity as a vectorial average among the samples. This technique was successfully employed to model the aerodynamics of vertical axis turbines (Schito et al., 2018; Melani et al., 2019). Thanks to EVM, the smearing parameter of the Regularization Kernel function (a bi-variate normal distribution) has been set equal to the characteristic cell size without problems of numerical stability. The length and position of the sampling
line has followed the optimal values reported in Schito and Zasso (2014). Furthermore, the code has provided the possibility of imposing a surge motion to the actuator lines for replicating the unsteady wind tunnel tests. The airfoil polars that have been used for AL are the same adopted in Aero Module simulations and the spanwise chord and twist distributions have followed



the scaled model's specifics. Only the three blades have been considered (as rigid actuator lines), neither the tower nor the nacelle have been taken into account.

The computational domain has reproduced faithfully the wind tunnel section's width and height. The streamwise direction has been modified setting the inlet section 5 m upstream of the turbine, i.e. where the wind velocity was measured, and the outlet more than six diameters downstream, to allow the atmospheric pressure recovery. The walls have been assumed smooth to avoid the need of modelling the boundary layers. Thanks to the absence of the turbine, a completely structured and incoming flow aligned grid has been generated. Cubic elements have been used in the rotor zone and two cylindrical refinement zones

have been set around the turbine. The detailed grid layout can be found in Mancini (2020). The chosen mesh has almost 3.5 millions of cells, with 50 elements per actuator line. Using a finer grid (11.6 millions elements, 75 per blade line) with the same layout, the average steady turbine loads heve varied of less than 1 %.

Thanks to the absence of boundary layers, Large Eddy Simulations (LES) have been conducted to solve the incompressible Navier-Stokes equations, featuring the standard Smagorinsky model. More complex sub grid scale models could have been

selected but Sarlak et al. (2015) proved their impact small, provided that sufficient grid refinement is present. A third order QUICK scheme has been used for the convective term, with an almost purely second order Crank-Nicolson scheme for the time derivatives. The solver is based on the PISO algorithm, using a multi grid linear solver for pressure and a preconditioned bi-conjugate gradient method for the velocity components. The time step size ($\Delta t = 0.0005$ s) has been selected in order to: keep the Courant number below 0.5; prevent actuator line tips from crossing more than one cell per time step; avoid leakage in

the frequency domain analysis.

### 3.3    CFD

The fully-resolved CFD simulations have been run for a subset of cases to get more insights into the flow physics. The finite-volume flow solver FLOWer, originally developed by the German Aerospace Center, has been used for the present study (Kroll and Faßbender, 2005).

The computational setup of the one third model of the scaled wind turbine, presented in Cormier et al. (2018), has been extended to a full model of the wind turbine as represented in Fig. 4. The compressible unsteady Reynolds-Averaged Navier-Stokes equations have been solved, using the Menter's shear stress transport model to model the turbulence (Menter, 1994). A second order dual time-stepping scheme has been used for the time discretization and a multigrid algorithm has been applied to accelerate the convergence. The $5^{th}$ order Weighted Non Oscillatory (WENO) scheme has been used for spatial discretization

in the wake of the wind turbine, in order to reduce the dissipation of the vortices (Kowarsch et al., 2013). In the body meshes and outside the wake region, spatial discretization has been realised with the $2^{nd}$ order Jameson-Schmidt-Turkel (JST) scheme (Jameson et al., 1981). All body grids have been embedded in a Cartesian background mesh thanks to the Chimera grid overlapping technique. The hub has been extended from a 120 ° to a 360 ° section and new meshes for the tower and its base have been generated. The grids have been generated with the commercial tool Pointwise, combined with in-house automatising

scripts. The height of the first boundary layer cell in the body meshes has been chosen such that $y^+ \approx 1$ is ensured. The resulting numerical setup consists of 118 Mio. cells. As the experimental streamwise velocity profile upstream the turbine presented no





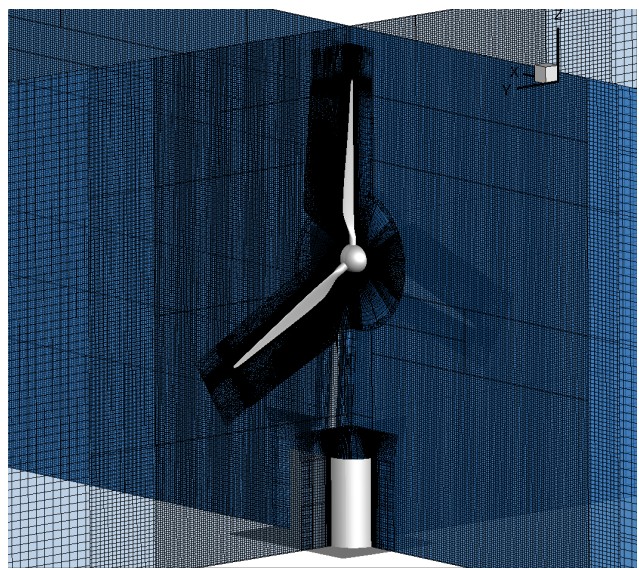

**Figure 4.** Numerical setup of the fully-resolved wind turbine (CFD).

shear in the rotor area, a uniform inflow has been applied at the inlet via a far-field boundary condition. To take into account the blockage effect of the wind tunnel's upper and lower walls while optimizing the use of computational resources, the ceiling and ground have been modeled by a slip boundary condition, taking care to add a displacement thickness of $12.5\,\mathrm{cm}$ to meet the

experimental flow rate. The distance between the wind turbine and the boundaries of the computational box has been defined according to Sayed et al. (2015), who studied the influence of the distance to boundaries on the wind turbine aerodynamics. The outlet and lateral boundary conditions have been set as far-field too and located respectively 9 and 5 rotor radii away from the wind turbine. A time step corresponding to a blade rotation of $1\,^{\circ}$ with 60 inner iterations has been applied.

## 4  Results

Since dynamic inflow effects are known to be more relevant when the turbine loading is high, it has been decided to focus the comparison on rated wind conditions rather than above-rated. Furthermore, RATED2 tests (see Table 2) have been preferred to RATED1 because of the better signal to noise ratio characterizing the measurements.

### 4.1  Steady comparison

The steady performances of the scaled turbine are considered first, comparing the predictions of the different codes against the

experiments without surge. The outcomes of this comparison are reported in Table 3, in terms of steady thrust force ($T_0$) and mechanical power ($P_0$); the percentage errors have been defined with respect to wind tunnel measurements. To run the steady CFD simulations, only the axisymmetric model (called 1/3 CFD) has been used. However, the good agreement found with the quasi-steady theory (Sect. 4.2), has given trust on estimating the steady performances from the full CFD model (called CFD





**Table 3.** Comparison of the steady turbine model performances in RATED2 conditions.

|  | **WT** | **CFD** | **AL** | **1/3 CFD** | **AWSM** | **BEM** |
|---|---|---|---|---|---|---|
| $T_0$ **[N]** | 35.91 | 36.57 | 36.60 | 34.20 | 35.00 | 34.65 |
| error $T_0$ | / | +1.84 % | +1.92 % | -4.76 % | -2.53 % | -3.51 % |
| $P_0$ **[W]** | 83.79 | 84.29 | 87.07 | 73.44 | 75.5 | 73.95 |
| error $P_0$ | / | +0.6 % | +3.92 % | -12.35 % | -9.89 % | -11.75 % |

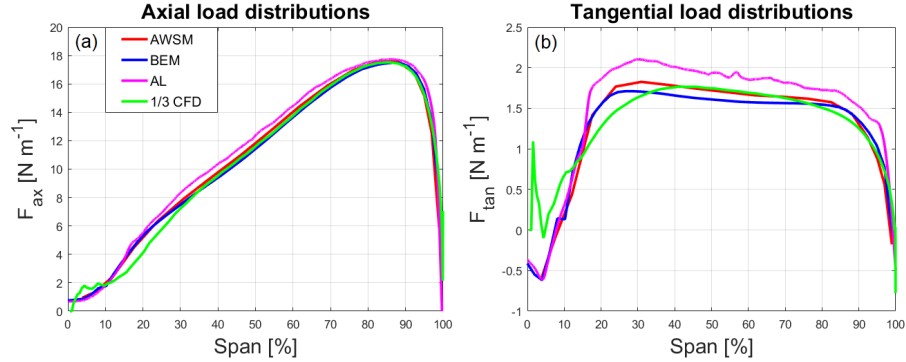

**Figure 5.** Axial and tangential spanwise loads distributions comparison.

in Table 3), averaging the unsteady loads over a full surge period. The consistency of this approach has been confirmed by the

excellent match with steady wind tunnel tests, showing maximum discrepancies below two percent. The confidence has been raised further by the fact that the average values obtained from the two different surge simulations are almost coincident.

Similarly to the full CFD, also the actuator line results are in very good agreement with experiments. The Aero-Module codes (BEM and AWSM) show more relevant discrepancies instead, especially for the power (i.e. torque) that is underestimated of about 10 % by both models. The thrust is underrated as well, but to a lower extent. Very similar values have been obtained

by the axisymmetric CFD simulation too; hence BEM, AWSM and 1/3 CFD results are in good agreement among each other, but systematically different than full CFD, AL and experimental tests. A significant source of this discrepancy appeared to be a slight difference in inflow velocity due to the fact that the reference wind speed in wind tunnel tests was measured 5 m upstream of the rotor, where the induction field had indeed an impact, albeit small, and this was not accounted by all the models. However, the influence of such discrepancy on the unsteady investigation is expected to be negligible.

To deepen the steady comparison, the spanwise loads distributions obtained with the different codes have been considered. In Fig. 5 the axial and tangential, i.e. contributing to thrust and torque respectively, unit forces distributions along the span are reported. Unfortunately, the spanwise distributions from CFD have been extracted from the steady axisymmetric simulation only. For what concerns the the axial load (Fig. 5a), the shape is the same for all the models and the discrepancies are small throughout the whole span. In accordance to the integral values the match among BEM, AWSM and 1/3 CFD is almost perfect;





AL's distribution instead, is just slightly above the others. A greater discrepancy is found for the tangential force (Fig. 5b).
Here, the shapes of AL, BEM and AWSM distributions are very similar to each other (owing to the same polars used), but
the first shows greater values from 25 % of the span on. The lower values besides similar overall shapes confirm the impact
of the rotor induced velocity on the measured wind speed ahead of the turbine. A greater undisturbed velocity would indeed
increase the angle of attack along the span, leading to higher values of axial and tangential forces with the same distributions'
shapes, similarly to AL. Because of the presence of the nacelle, the shape of the 1/3 CFD differs significantly from the others
at the blade root, aligning to them only around 40 % of the span. The root discrepancy does not produce any significant power
variation though, since its contribution to the integral torque is small.

### 4.2    Unsteady comparison

After having validated the code predictions in the stationary turbine case, some of the surge tests belonging to the UNAFLOW
matrix have been considered, always in RATED2 conditions (Table 2). The list of these tests is shown in Table 4, where the
surge parameters are given (in model scale) along with the corresponding wind tunnel test number. The only tests replicated
by all the codes have been numbers 50 and 59.

The primary target of the unsteady experimental campaign was the characterization of the thrust force oscillation, due to
its leading role in the surge dynamics of a FOWT. Indeed, the scaled model was specifically designed to match the RWT
thrust coefficient. Anyway, having the quantity available from both codes and experimental measurements, also the mechanical
power has been taken into account. However, wind tunnel torque measurements have been discovered affected by a mechanical
resonance that biased the high frequency results. For this reason, the analysis hereinafter presented is mostly focused on the
thrust. Concerning the power, only the comparison of the surge frequency harmonic is shown, in Sect. 4.2.2, for completeness
sake.

**4.2.1    Time domain analysis**

The comparison of the thrust oscillation is first presented in time domain, as typically found in literature. As starting point of
the analysis, the impact of the surge motion on the mean aerodynamic thrust has been assessed, since Micallef and Sant and
Farrugia et al. observed a variation of the mean thrust coefficient during surge, also at the optimal $\lambda$. To check if the results are
characterized by a similar behaviour, a mean thrust variation parameter can be defined as:

$$\epsilon_T = 1 + \frac{\overline{T} - T_0}{T_0} \; ; \tag{4}$$

being $T_0$ the steady value reported in Table 3 and $\overline{T}$ the average of the thrust signal over a full surge period. Figure 6 plots
the values of $\epsilon_T$, from the different tests and simulations performed, against the surge frequency. In all the cases considered
the surge motion seems not to affect the mean thrust anyhow. The maximum discrepancies with respect to the steady values
are always below 0.5 % and utterly insensitive to the surge parameters. Such small variations fall within the uncertainty level
associated to each method. Therefore, for the purpose of this work it is possible to consider $T_0 \cong \overline{T}$.



**Table 4.** Numerical-experimental tests matrix: * exp + AL; ** exp + BEM + AWSM; *** exp + all codes.

| UNAFLOW # | $f_s$ [Hz] | $A_s$ [mm] |
|-----------|-----------|-----------|
| **33\*\*** | 0.125 | 125 |
| **37\*** | 0.25 | 125 |
| **41\*** | 0.5 | 65 |
| **45\*** | 0.75 | 40 |
| **49\*** | 1 | 50 |
| **50\*\*\*** | 1 | 35 |
| **51\*** | 1 | 25 |
| **53\*** | 1.5 | 20 |
| **55\*** | 1.5 | 10 |
| **57\*** | 2 | 15 |
| **59\*\*\*** | 2 | 8 |

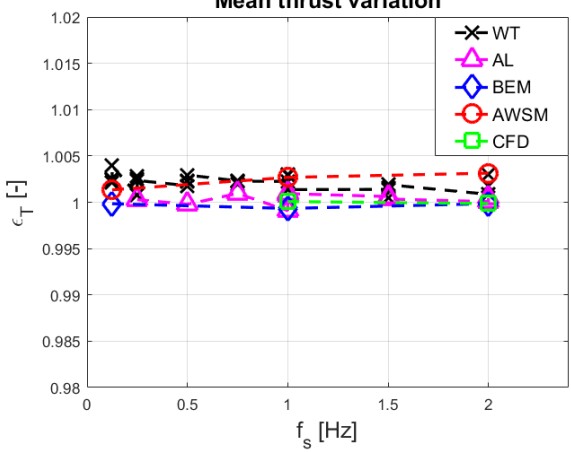

**Figure 6.** Mean thrust variation with the surge frequency.

To continue the time domain thrust analysis, it has been decided to separate the unsteady part of the signals from the steady one, subtracting the mean values from the thrust time histories. Thus considering:

$$\Delta T(t) = T(t) - T_0 . \tag{5}$$

This way it is possible to avoid the steady discrepancies when comparing the time histories of different methods. In Fig. 7
the time histories of $\Delta T$, obtained by the different codes, are compared to the experimental measurement and to the linear quasi-steady model prediction (Appendix A). The plot refers to the unsteady test number 59, which has been reproduced with all the codes (Table 4), but the comments apply for the other tests as well. From Fig. 7a it is evident that the nacelle balance

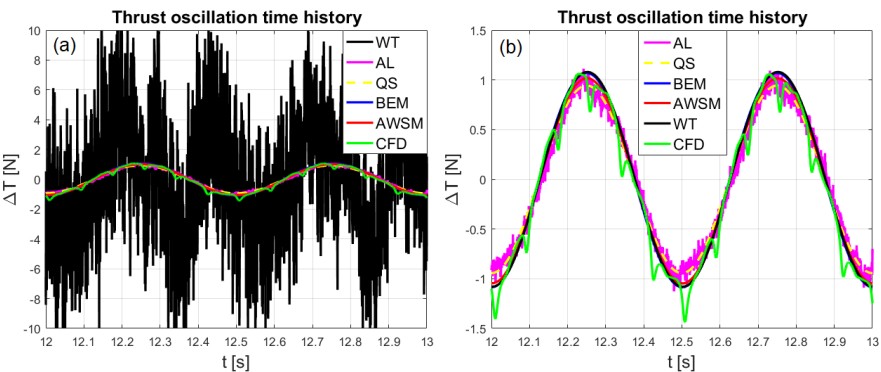

**Figure 7.** Thrust oscillation time histories for case 59 (a); same plot considering only the surge harmonic of wind tunnel measurements (b).

measurement was characterized by a relevant amount of disturbances, mainly caused by mechanical vibrations (e.g. platform's high frequency dynamics, imperfect surge actuation, rotordynamic effects, etc.), aerodynamic turbulence and instruments noise.

Also the inertia subtraction procedure has contributed to the presence of such high frequency components, since it is effective only at the surge harmonic. This testifies the complexity associated with the experimental investigation of the aerodynamics of a turbine subjected to imposed motions. To solve the problem, an harmonic filtering procedure has been followed. This way the wind tunnel signals have been cleaned extracting only the surge frequency harmonic, getting rid of all the spurious effects. Thanks to the long experimental observation periods indeed, it has been possible to get high resolution spectra despite the high

sampling rate. In addition, leakage has always been avoided considering time windows lengths integer multiples of the test's surge period.

The output of this cleaning procedure, applied to the thrust oscillation of test 59, is reported in Fig. 7b. The comparison reveals a very good agreement among codes, wind tunnel measurement and quasi-steady theory, confirming the effectiveness of extracting the surge harmonic from the experimental data. In fact, the different codes predictions appear totally dominated

by the surge frequency component. In particular, BEM and AWSM responses are almost purely mono-harmonic. In AL large eddy simulations instead, a certain amount of high frequency components is noticeable, but insignificant with respect to the surge harmonic. The main cause of these oscillations is the turbulence that, albeit weak because of the smooth flow boundary condition at inlet, forms upstream the turbine because of both the high wind tunnel Reynolds number ($\sim 1.6 \cdot 10^6$) and the influence of the actuator forces in the rotor plane.

Finally, the full CFD signal presents a clear component also at the blade passing frequency, due to the modelling of the turbine tower. Qualitatively, the assessment of the unsteady time histories shows a promising agreement, with the responses that often overlap with each other. Nevertheless, the time domain analysis hinders a quantitative comparison among the codes, the experiments and the quasi-steady theory because the differences are small enough to be hardly recognizable.



### 4.2.2 Frequency domain analysis

Having observed that the surge harmonic rules the aerodynamic response of the turbine, its frequency domain characterization becomes fundamental to validate the results. This way the unsteady response is completely described by its amplitude and phase and, thanks to the clear reference provided by the quasi-steady theory (Appendx A), it becomes much easier to spot dynamic inflow effects due to surge. Indicating now with $\Delta T$ only the surge harmonic of the thrust oscillation, it is possible to represent it in the complex plane as:

$$\Delta T = |\Delta T|\,e^{i\phi} = |\Delta T|\,(\cos\phi + i\sin\phi)\;; \tag{6}$$

being $\phi$ the phase shift between the thrust oscillation and the surge displacement at the surge frequency and $i$ the imaginary unit. For wind tunnel measurements, the base displacement signal imposed by the surge actuator, Eq. (1), has been chosen as phase reference. The resulting scheme is reported in Fig. 8a. For control purposes, the thrust oscillation harmonic at the surge frequency can be more conveniently expressed in terms of the states of the system, defining two coefficients of utmost

importance for surge stability assessment: the aerodynamic damping ($c_{aero}$) and the aerodynamic mass ($m_{aero}$). Therefore, $\Delta T$ can be expressed in terms of these parameters as:

$$\Delta T = -c_{aero}\dot{x} - m_{aero}\ddot{x}\;; \tag{7}$$

with:

$$\dot{x} = i\,2\pi f_s\,A_s\;; \tag{8}$$

$$\ddot{x} = -(2\pi f_s)^2\,A_s\;. \tag{9}$$

Combining Eq. (7), (8) and (9) the expressions for the aerodynamic damping and mass coefficients are immediately derived:

$$c_{aero} = -\frac{|\Delta T|\cos\phi}{2\pi f_s\,A_s}\;; \tag{10}$$

$$m_{aero} = \frac{|\Delta T|\sin\phi}{(2\pi f_s)^2\,A_s}\;. \tag{11}$$

In order to extend the generality of the results obtained, paving the way for a more robust comparisons also in future work, a

non-dimensional characterization of the thrust oscillation harmonic at the surge frequency is proposed. For the purpose, a few non-dimensional groups have been defined. The first two are required to characterize the surge motion and have been called the *surge reduced frequency* ($f_{red}$) and the *surge reduced amplitude* ($A_{red}$) respectively. They are defined as:

$$f_{red} = \frac{f_s\,D}{V_0}\;; \tag{12}$$

$$A_{red} = \frac{A_s}{D}\;; \tag{13}$$

being $D$ the turbine diameter and $V_0$ the free stream wind velocity. Note that the reduced frequency is the inverse of the reduced velocity defined by Bayati et al. (2017b), and it compares the frequency of surge to that associated with dynamic inflow, which





is the most relevant source of unsteadiness associated to floaters' motions. The higher $f_{red}$, the greater the chance that dynamic inflow will affect the response. The reduced amplitude instead, might be used to evaluate the validity boundaries of the small displacements assumption required to get the linear quasi-steady model (Appendix A). To fully characterize the surge harmonic

of the thrust response, its phase has been used directly, whilst for the amplitude an *unsteady thrust coefficient* ($C_{\Delta T}$) has been defined following the steady thrust coefficient definition:

$$C_{\Delta T} = \frac{|\Delta T|}{0.5\,\rho\,A_D\,V_0^2}\;; \tag{14}$$

being $\rho$ the air density and $A_D$ the area of the disk swept by the blades. Relying on the quasi-steady assumption, Eq. (A5) can be reworked letting non-dimensional groups to appear so that an expression for the unsteady thrust coefficient is found:

$$C_{\Delta T} = 2\pi\,c_0^*\,f_{red}\,A_{red}\,. \tag{15}$$

The coefficient $c_0^*$ has been derived from the nondimensionalization of Eq. (A7) and it has been called *non-dimensional steady aerodynamic damping*. The interesting fact is that it is only a function of the steady thrust coefficient curve of the turbine $C_T(\lambda)$, in fact:

$$c_0^* = \frac{c_0}{0.5\,\rho\,A_D\,V_0} = 2\,C_T(\lambda_0) - \frac{dC_T}{d\lambda}|_{\lambda_0} \cdot \lambda_0\;; \tag{16}$$

being $\lambda_0$ the steady operating conditions' tip speed ratio and $c_0$ the steady aerodynamic damping defined in Appendix A.

Exploiting the new variables, the results comparison is presented in non-dimensional form. In Fig. 8b the amplitude of the thrust oscillation at the surge harmonic is characterized, plotting the ratio between the unsteady thrust coefficient and the reduced surge amplitude against the surge reduced frequency. The reason behind this choice is the linear trend foreseen by the quasi-steady theory, i.e. Eq. (15), that provides a clear theoretical reference for the comparison. It is worth to underline

that the slope of the quasi-steady reference has been evaluated analytically using the RWT characteristic curve, as explained in Appendix A. The plot reveals an excellent agreement among all the codes involved and wind tunnel tests, with a maximum deviation around 10 % at the highest reduced frequency. Anyway, all the numerical predictions fall inside the experimental tests scatter. Diversely from the steady turbine case, BEM, CFD and AWSM tend to predict slightly higher values than AL and analytical model, with wind tunnel measurements typically in between. Although, all the data seem to confirm the linear trend

predicted with the quasi-steady assumption. The comparison in terms of phase of $\Delta T$ is shown in Fig. 8c. According to the reference system of Fig. 2b, the quasi-steady model foresees $\Delta T$ to be in opposition of phase with respect to the surge velocity. Having referred the phase to the surge displacement, the reference value is then $\phi = -90\,°$. Once again the codes agree closely with the quasi-steady theory, with discrepancies slightly increasing with $f_{red}$. The phase values from wind tunnel tests instead, show a relevant scatter because of the uncertainty entailed by the inertia subtraction procedure. Especially at high frequencies

indeed, the share of aerodynamic thrust in the balance measurement is much smaller than the inertial contribution due to surge acceleration and, when the subtraction is performed, the phase of $\Delta T$ appears much more sensitive to disturbances than its amplitude (Mancini, 2020).

Knowing amplitude and phase of the thrust oscillation's surge harmonic, it is possible to evaluate the aerodynamic mass and damping coefficients from Eq. (10) and (11). To continue with a non-dimensional analysis, the *non-dimensional aerodynamic*

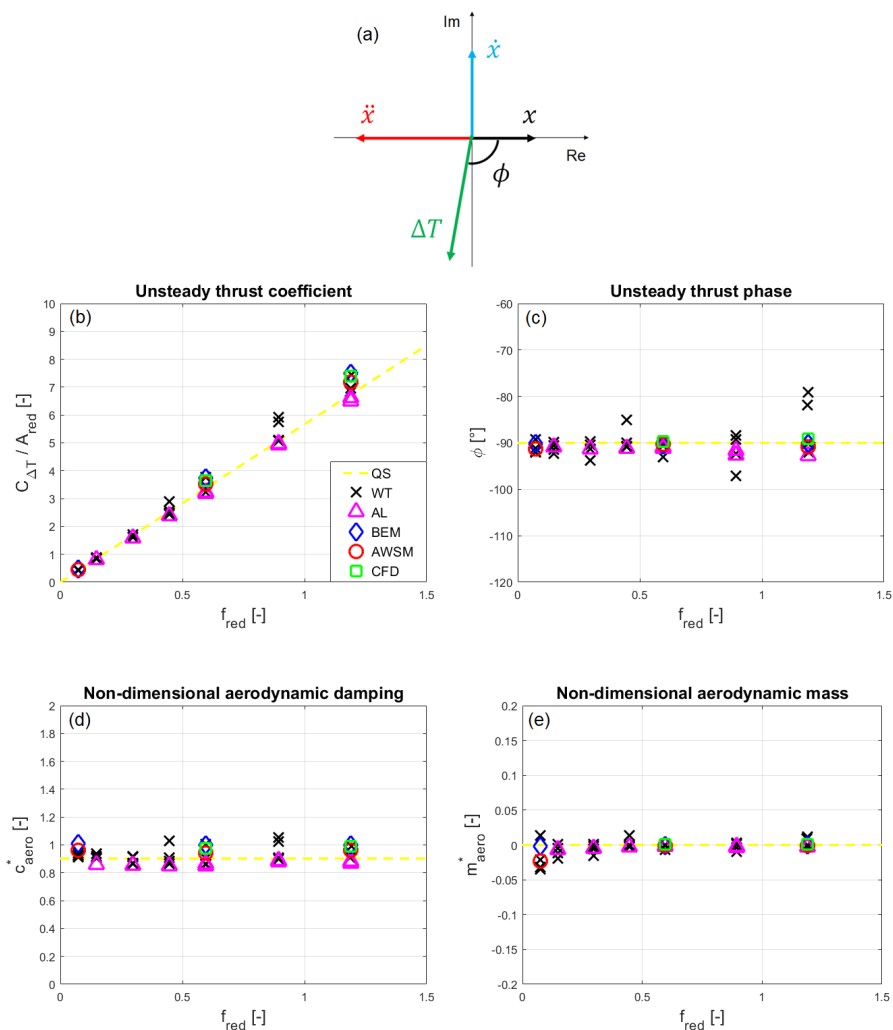

**Figure 8.** Complex representation of the thrust oscillation's surge harmonic (a); unsteady thrust coefficient comparison (b); comparison of the phase of the of the thrust oscillation's surge harmonic (c); aerodynamic damping comparison (d); aerodynamic mass comparison (e).

*damping* ($c^*_{aero}$) and the *non-dimensional aerodynamic mass* ($m^*_{aero}$) have been defined as:

$$c^*_{aero} = \frac{c_{aero}}{0.5\,\rho\,A_D\,V_0} \; ; \tag{17}$$

$$m^*_{aero} = \frac{m_{aero}}{\rho\,A_D\,D} \; . \tag{18}$$

According to the quasi-steady theory $c^*_{aero} = c^*_0$ and $m^*_{aero} = 0$. The non-dimensional comparison in terms of aerodynamic damping is reported in Fig. 8d. All the codes show a constant trend with respect to the reduced frequency, confirming the

linearity of the plot in Fig. 8b, thus the validity of the quasi-steady assumption. Concerning the non-dimensional aerodynamic

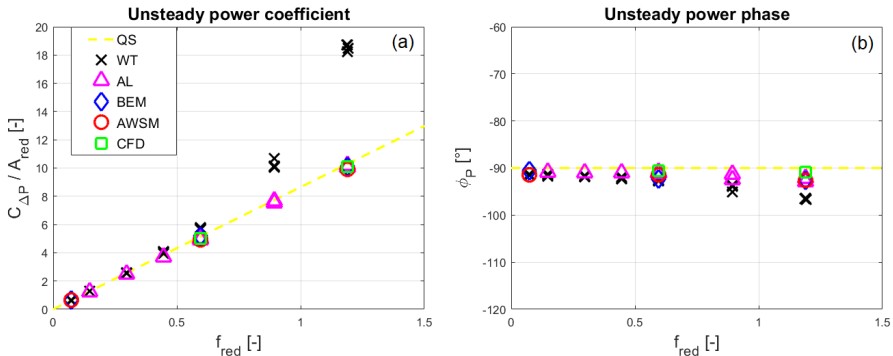

**Figure 9.** Unsteady power coefficient comparison (a); comparison of the phase of the of the power oscillation's surge harmonic (b).

mass, Fig. 8e confirms that its values are always extremely close to zero, in agreement with the quasi-steady theory. Only a slight scatter appears at the lowest frequencies because of the inverse dependency of $m_{aero}$ on the square of $f_s$, as in Eq. (11); this amplifies even very small phase errors, leading to unphysical values of the aerodynamic mass.

Very similar considerations to those regarding the thrust can be made for the power oscillation. In particular, being the unsteady response dominated by the surge harmonic, it is convenient to characterize it frequency domain. Similarly to the thrust, it is possible to represent the power oscillation's surge harmonic ($\Delta P$) in the complex plane as:

$$\Delta P = |\Delta P| e^{i\phi_P} = |\Delta P| (\cos\phi_P + i\sin\phi_P) \ ; \tag{19}$$

being $\phi_P$ the argument of $\Delta P$ always with respect to the surge displacement harmonic. Differently from the thrust case, the expression of the power oscillation in terms of the sytem's states is avoided since $\Delta P$ does not affect the stability, but only the

power harvesting. Passing in non-dimensional form, it is possible to define the *unsteady power coefficient* ($C_{\Delta P}$) as:

$$C_{\Delta P} = \frac{|\Delta P|}{0.5\,\rho\,A_D\,V_0^3} \ . \tag{20}$$

Then, reworking Eq. (A6), a non-dimensional expression linking the unsteady power coefficient to the steady turbine's operating conditions and to the surge parameters can be found, again relying on the quasi-steady assumption:

$$C_{\Delta P} = 2\pi\,\zeta_0^*\,f_{red}\,A_{red} \ . \tag{21}$$

Latter expression perfectly corresponds to Eq. (15) concerning the thrust. Only this time the parameter depending on the turbine's steady performance is $\zeta_0^*$ rather than $c_0^*$, and it is defined as:

$$\zeta_0^* = \frac{\zeta_0}{0.5\,\rho\,A_D\,V_0^2} = 3\,C_P(\lambda_0) - \frac{dC_P}{d\lambda}\big|_{\lambda_0}\cdot\lambda_0 \ ; \tag{22}$$

being $C_P$ the turbine's power coefficient and $\zeta_0$ the parameter defined in Eq. (A8), which links the power oscillation to the surge velocity.





The comparison in terms of unsteady power coefficient is reported in Fig. 9a, always dividing by the reduced surge amplitude and plotting against the reduced frequency to have a linear quasi-steady reference. As previously anticipated, the torque measured by the balances in wind tunnel tests was subjected to a dynamic effect altering the power oscillation in the higher frequency cases. In fact, the sharp amplitude increase, arising as soon as $f_s$ exceeded 1 Hz, and the contextual phase reduction (Fig. 9b) were caused by a powertrain resonance standing at 3.95 Hz. As long as the natural frequency was far, the angular degree of freedom behaved quasi-statically with respect to such vibration mode and the results were almost unaffected; getting closer to the resonance, the typical mechanical amplification phenomenon occurred. As a result, only low $f_{red}$ cases have been validated by wind tunnel measurements. However, the excellent agreement among all the codes and the quasi-steady model gives a great confidence on the numerical results' validity regarding $\Delta P$ for the whole frequency range. The quasi-steady behaviour found is also totally coherent to what observed for the thrust, in which the codes predictions have been confirmed by the experiments. If new unsteady tests will be conducted, some stiffness will be added to the angular degree of freedom (e.g. changing the transmission belt) in order to move resonance farther from the $f_s$ range considered.

The phase comparison is reported in Fig. 9b and confirms the conclusions of the unsteady power coefficient case. Leaving the wind tunnel measurements aside, the codes show little discrepancies among each other. Nevertheless, a reduction of $\phi_P$ with respect to the quasi-steady value appears to occur at the higher frequencies, resembling a dynamic inflow effect. A similar reduction has also been found by actuator line in the phase of the thrust, with the other codes showing values closer to $-90\,°$. In the power case, the codes seem to be more concordant in the presence of this slight delay. However, no matter which code is considered, the maximum phase shift with respect to the quasi-steady reference is always below three degrees thus negligible. As for the amplitude plot, low frequency wind tunnel tests utterly confirm the numerical outcomes, whilst a phase shift due to the resonance affects the higher $f_{red}$ results.

# 5 Conclusions

The performance response to an harmonic surge motion of a 1:75 scaled version of the DTU10MW RWT has been investigated using state of art numerical models with different fidelity levels. For the first time, the unsteady results have been validated against high fidelity wind tunnel tests specifically focused on the aerodynamics. These tests, in which the surge motion was imposed to the turbine, were conducted in Polimi's facility (GVPM) within the UNAFLOW project. The comparison has revealed a surprisingly good agreement among the different codes' predictions, with smaller discrepancies in the unsteady case than the steady one. The codes have all confirmed the aerodynamic response to be dominated by the component at the surge frequency. Hence, considering only that harmonic, it has been possible to clean the experimental measurements that were characterized by significant disturbances due to the unsteady tests' complexity. The resulting thrust measurements have validated the codes' predictions for the whole test matrix. Concerning the torque instead, the experiments have been able to confirm only the low frequency outcomes, since the higher frequency signals were biased by a mechanical resonance. However, the excellent numerical results' agreement suggests the validity of the codes' predictions also for the unsteady power.





Owing to its leading role in the aerodynamic response, the surge harmonic has been characterized in frequency domain. This has allowed to perform a more quantitative comparison of the unsteady results, at the same time focusing on control relevant parameters. The analysis has been presented in non-dimensional form aiming to maximize its generality. The focus on the surge
harmonic has given the possibility to define a linear analytical model, based on the quasi-steady assumption, with which both numerical and experimental results have been further validated. Despite the several approximations made, the quasi-steady model has shown an outstanding match with the other data, allowing to confirm the conclusions drawn by de Vaal et al. (2014), so that the aerodynamic response of a floating wind turbine at rated wind conditions to typical wave induced surge motions can be well modelled relying on the quasi-steady assumption. In the conditions considered indeed, rotor unsteadiness has had
little influence on the loads and even the BEM code has produced accurate results using a classical dynamic inflow model. The absence of mean performance variations due to surge has been an ulterior proof corroborating this evidence. Nevertheless, such conclusion is tightly liked to the frequency range selected as well as to the specific time scale of dynamic inflow. In fact, $f_{red}$ is the parameter that rules the impact of dynamic inflow effects. In this work its values have not exceeded 1.2, but the increasing results scatter towards higher frequencies likely indicates the inception of unsteady effects. The results presented have revealed
that the accuracy of the quasi-steady assumption is almost insensitive to the surge reduced amplitude. Although, it should be verified up to what threshold non-linear effects can be neglected and the small displacements assumption holds. Anyhow, the size of the rotors currently employed in offshore wind farms warrants little concern about the magnitude of $A_{red}$.

The linear quasi-steady model proposed, expressed in non-dimensional terms, might be a convenient tool also for future work. As long as a similar reduced frequency range is considered indeed, the loads oscillation amplitudes can be effectively
estimated by means of Eq. (15) and (21), whilst the phase can be reasonably assumed equal to the quasi-steady reference. This approach separates the influence of the surge parameters from that of the steady operating conditions, allowing to better understand the impact that each single variable has on the unsteady behaviour. Furthermore, its integral load perspective makes it suitable for control strategy design and assessment. For example, the increase of the loads oscillation amplitudes found by Micallef and Sant (2015) raising the tip speed ratio at constant $A_s$ and $f_s$, may be explained by an increase of both $c_0^*$ and
$\zeta_0^*$ linked to the steady characteristic curves' shapes that, of course, depend on the controller. Moreover, the critical operating points where the stability is in jeopardy because of small (or negative) aerodynamic damping can be immediately found from the expression of $c_0^*$. Then, the control strategy can be adjusted to modify the steady characteristic curves, adding some more surge damping where needed. In fact, a higher $c_0^*$ means higher $C_{\Delta T}$ only if surge is assumed imposed in Eq. (15); in reality a higher damping would drastically reduce $A_{red}$, providing a benefit overall.

In future work, higher reduced frequency cases where dynamic inflow effects appear will be addressed to understand what happens when the quasi-steady assumption falls. The codes validation effort hereby described has increased the confidence on the numerical predictions indeed, paving the way for considering more critical cases. A similar characterization will be also attempted for the turbine pitch case, which is expected to be more challenging due to the radial variation of the imposed motion. Finally, a revision of the powertrain assembly is being carried out to make sure that, if new unsteady experiments had
to be conducted, the torque measurements would not be affected by any resonance.



*Data availability.* All the data presented in this work are stored in a FTP server, together with the whole UNAFLOW database. Upon request the access keys will be granted out of charges to anyone interested.

## Appendix A: Linear quasi-steady model

Exploiting the quasi-steady assumption, it is possible to obtain a theoretical reference for the unsteady performances of a turbine
subjected to surge. In fact, as long as the motion period is long compared to the time scale of dynamic inflow, i.e. the reduced frequency of Eq. (12) is small, the induction field can be assumed to adjust immediately to the relative wind change imposed by surge. If dynamic stall effects are neglected, the hypothesis of no dynamic inflow automatically implies the absence of airfoil unsteadiness, since it occurs at a shorter time scale. Thus, assuming a quasi-steady behaviour, the turbine performances can be expressed in terms of thrust and power coefficients and the surge motion reduces to a change of the incoming wind speed
experienced by the rotor ($V_w$). In particular:

$$V_w = V_0 - \dot{x} \; ; \tag{A1}$$

having used the reference system of Fig. 2b. This modifies the expression of the tip speed ratio that becomes:

$$\lambda_w = \frac{\Omega D}{2\,V_w} \; . \tag{A2}$$

Consequently, the turbine's thrust and power responses can be expressed as:

$$T = \frac{1}{2}\,\rho\,A_D\,C_T(\lambda_w)\,V_w^2 \; ; \tag{A3}$$

$$P = \frac{1}{2}\,\rho\,A_D\,C_P(\lambda_w)\,V_w^2 \; . \tag{A4}$$

To obtain the easy reference used in the paper, the expressions have been linearized for small surge velocities, i.e. $\dot{x} \rightarrow 0$. In case of harmonic surge displacement, this can be translated to a condition on the reduced surge amplitude $A_{red} \rightarrow 0$, which means $A_s << D$. Hence, the linear approximation is likely to be suitable for modern multi-megawatt rotors employed in
floating wind farms. Considering small variations around the steady operating conditions and a constant rotational speed (as in wind tunnel tests) the following expressions for the thrust and power oscillations have been obtained:

$$\Delta T \approx -c_0\,\dot{x} \; ; \tag{A5}$$

$$\Delta P \approx -\zeta_0\,\dot{x} \; ; \tag{A6}$$

with $c_0$ and $\zeta_0$ functions only of steady operating conditions of the turbine, defined as:

$$c_0 = -\frac{dT}{d\dot{x}}|_{\dot{x}=0} = \frac{1}{2}\,\rho\,A_D\,[2\,V_0\,C_T(\lambda_0) - \frac{dC_T}{d\lambda}|_{\lambda_0}\,\frac{\Omega D}{2}] \; ; \tag{A7}$$

$$\zeta_0 = -\frac{dP}{d\dot{x}}|_{\dot{x}=0} = \frac{1}{2}\,\rho\,A_D\,V_0\,[3\,V_0\,C_P(\lambda_0) - \frac{dC_P}{d\lambda}|_{\lambda_0}\,\frac{\Omega D}{2}] \; . \tag{A8}$$

By means of this simplified approach it is possible to estimate the unsteady response knowing the steady operating point, the characteristic curves and the surge motion parameters. Provided that the scaled model's complete characteristic curves





were unavailable, those of the RWT have been used. The scaled turbine was designed to match the DTU10MW RWT thrust
coefficient indeed, but also the power coefficient was well reproduced in rated conditions (Bayati et al., 2017a). Although, the
RWT performance curves take into account also the regulation, whilst in the experimental campaign both the rotational speed
and the blades pitch were kept constant. To bypass this issue the shapes of the curves in the neighbourhood of $\lambda = 7.5$ have
been approximated taking three points where the regulation has little or none influence, fitting them with a quadratic trend.
Except from that at optimal tip speed ratio, the other two points have been selected as close as possible to the first, but towards
higher $\lambda$ (i.e. $\lambda = 8$ and 9.3). In the below rated region, not too far from $\lambda = 7.5$, the pitch regulation is very small in fact and
the rotational speed stays constant at the minimum value. Such procedure had to be followed for evaluating $c_0$, whilst for $\zeta_0$
the derivative of $C_P$ at the optimal tip speed ratio is obviously close to zero and the knowledge of $C_P(\lambda_0)$ is enough. Despite
its simplicity, this approach can provide accurate predictions as long as: the quasi-steady assumption holds, i.e. $f_{red} \rightarrow 0$ thus
$f_s << V_0/D$; the surge velocity is small, i.e. $A_{red} \rightarrow 0$ thus $A_s >> D$; the right characteristic curves are used, i.e. if the
regulation is active during surge the curves have to take it into account or vice versa. Finally, it is worth to notice that also a
variable rotational speed might be considered adding little complication to the model.

*Author contributions.* S. Mancini conducted the UNAFLOW project revision, run the actuator line simulations and coordinated the manuscript
composition under the supervision of A. Zasso and P. Schito. K. Boorsma and M. Caboni performed BEM and AWSM simulations, while
M. Cormier and T. Lutz provided the CFD results. All the authors contributed with their ideas and thoughts to the development of the paper.

*Competing interests.* The authors declare that they have no conflict of interest.

*Acknowledgements.* This research has been funded by EU-EERA (European Energy Research Alliance)/IRPWIND Joint Experiment 2017.
The authors gratefully acknowledge HLRS Stuttgart and Cineca HPC for providing computational resources.



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
