# Peer review of "Characterization of the unsteady aerodynamic response of a floating offshore wind turbine to surge motion"

_Wind Energy Science, 2020_

## Referee Comment (RC1) · Anonymous Referee #1 · 4 Sep 2020

General comments: A downscaled model of the DTU10MW turbine is investigated which is subjected to small harmonic surge oscillations representing one possible motion of a floating wind turbine. Numerical models of different levels of fidelity are used to evaluate the thrust and power response of the turbine to the oscillation. The results are compared to experimental wind tunnel studies. The topic is of high relevance as the design of floating wind turbine is still affected by large uncertainties. The quality of the numerical and experimental methods is state of the art and the outcome is of great value for the community as a benchmark for code validation.

Specific comments: I would suggest precising the title to "Characterization of the unsteady aerodynamic response of a floating offshore wind turbine to surge motion"

Maybe is should be clarified early in the paper that whenever surge is mentioned,

harmonic surge is concidered.

Corrections:

Line 18: The variables in the equation should be introduced

Line 73: The bracket before "(Bayati et al., 2017b" should probably be moved "Bayati et al. (2017b"

Figure 2: Size should be increase for better readability

Line 160: "i.e. all what attached" -> "i.e. all what is attached"

Line 294: double "the"

Figure 7: Which test number or amplitude or shown?

Line 310: a comma behind "both" is missing

Line 347: This sentences is hard to understand and should be rephrased: "The main cause of these oscillations is the turbulence that, albeit weak because of the smooth flow boundary condition at inlet, forms upstream the turbine because of both the high wind tunnel Reynolds number (âĹij 1.6 Âů 106) and the influence of the actuator forces in the rotor plane"

Line 449: "...coherent to what was observed for the thrust..."

---

## Referee Comment (RC2) · Anonymous Referee #2 · 8 Sep 2020

In the paper the response of a 1:75 scaled version of the DTU 10MW RWT to platform surge motion is investigated. Numerical predictions obtained using different fidelity tools are compared to experimental results. The large amount of resources and work involved with this result is apparent. The non-dimensional approach proposed in the final part of the paper is a valuable contribution, although the robustness of the approach should be verified with other wind turbine rotors. I suggest that the article is accepted with minor revisions. I hope that the authors will address the "Major comments" in particular.

General Major Comments:

Figure 7a: In the raw WT timeseries a frequency double the surge frequency can be clearly seen. Please explain where this frequency originates and why it was filtered

out. This is quite important as the good agreement between numerical codes and experiments would not be achieved otherwise. Furthermore, this is crucial to justify the implicit assumption that thrust variations due to surge have the same frequency of the surge variation made in section 4.2.2. The filtering procedure should also be explained in further detail as the signal was filtered quite significantly.

Conclusions: "The codes have all confirmed the aerodynamic response to be dominated by the component at the surge frequency. Hence, considering only that harmonic, it has been possible to clean the experimental measurements that were characterized by significant disturbances due to the unsteady tests' complexity." – Could it be possible that the numerical models are not able to capture phenomena observed in the experiments? Please elaborate on this point.

English should be checked thoroughly. For instance, the preposition "a" and the article "the" are often missing or used inappropriately.

Specific Comments:

Introduction: The authors mention that the developed test rig has 2 degrees of freedom: pitch and surge. This study seems to be focused only on surge however, can you expain this choice in more detail? Furthermore, since the work focuses on basic understanding of the aerodynamic phenomena & code performance, it would be useful to the uninitiated if an explanation of the most aerodynamically relevant platform motions are and perhaps a diagram refencing those motions. At the very least authors should provide references for the mentioned information.

The introduction also mentions the lack of the influence of floating dynamics on WT control. In this paper however pitch control is disabled. I suggest shortening the paragraph and only mentioning that the results are presented in the frequency domain as well that is useful for controller design.

Section 2.2: I find the names RATED1 and RATED2 confusing. Although they both

refer to design TSR conditions, one refers to below-rated wind speeds. Controller behavior can be very different at rated and below rated. In accordance with the paper, I suggest the names to be changed to BELOW and RATED. I am open to other reasonable explanations.

Line 175: Please revise the phrase "Both models are lifting line codes, i.e. they make use of aerodynamic look-up tables to evaluate airfoil performance." The fact that a code uses aerodynamic look-up tables does not necessarily mean that it is a lifting-line code, actually BEM codes are typically not LL codes. If the specific code includes a lifting-line formulation for the blades and momentum modelling for the wake it should be clarified.

Section 3.1 Please clarify the effects that are being modelled with engineering models in BEM. This is crucial for a fair comparison.

Section 3.1.3 Have the values discussed here been validated by means of a sensitivity analysis? Other authors have suggested much shorter timesteps and longer wakes to obtain independent results.

Section 3.2 The authors mention that a LES simulation was performed. Was the Pope criterion or similar criteria to verify that an adequate percentage of the turbulent spectrum was resolved verified?

Section 3.3 How is the surge motion modelled in the CFD code? Please specify if automatic remeshing or grid deformation is applied or if there are rotating interfaces as sometimes seen when simulating rotors.

Figure 5: The authors might already be aware of this but it would be useful to include curves for the "full CFD" model a swell, to better highlight which model over-under estimates power & torque.

Table 4: Please clarify the parameters fs and As in the description

Figure 7: QS timeseries is had to make out, please choose another color

Section 4.2.2 It seems to me that in the formulas 6 and 7 the dependency from eˆ(2*pi*fs) was omitted. Please clarify this point. The same considerations apply to eq. 19.

---

## Short Comment (SC1) · 12 Sep 2020

Simone Mancini

simone1.mancini@mail.polimi.it

Dear referee,

thank you very much for your review, you raised some interesting points to discuss. I am working together with the co-authors to prepare a detailed response to all your remarks. It will take some time to conclude it, but I am pretty sure we will be able to deliver it within a couple of weeks.

For the moment, I can just clarify two minor points you mentioned:

1) I agree with you that the names RATED1 and RATED2 might be misleading and I also like the change you proposed (BELOW instead of RATED1). The names had been just inherited from the full turbine model test campaign conducted in Polimi; they had

not been changed to avoid discrepancies with the project report and especially with the FTP repository content. Therefore, I would propose to change the names to BELOW and RATED for the sake of clarity, although mentioning that they refer to RATED1 and RATED2 of the original database.

2) The lack of the steady force distributions from the full CFD model is indeed a pity, but unfortunately they were not included in the original project scope. Obtaining the curves from the raw CFD results would require additional budget and time resources that are currently unavailable. I am confident that this gap will be filled in future works.

Thank you again for the precious clues, you will soon receive a more detailed and complete reply.

Yours faithfully,

Simone Mancini
* * *

---

## Author Comment (AC1) · 12 Sep 2020

Dear referee,

thank you very much for your precious comments, tips and corrections on behalf of all co-authors.

Whether to include "surge motion" in the title has already been an object of discussion before submitting the preprint. Although we originally opted to exclude it for a leaner title, we agree that it makes it more in line with the contents of the paper and we will add it in the revised version as you suggested.

In the same way, we will try to specify a bit earlier that the focus is on harmonic surge.

Finally, concerning the proposed corrections we fully agree on all of them except for:

- Figure 7: the test case appears already specified in the caption as number 59 (whose parameters are reported in Table 4).

Thank you for your review.

Yours Faithfully,

Simone Mancini

---

## Author Comment (AC2) · 27 Sep 2020

Dear referee,

sorry for the late reply and thank you for your interesting remarks on behalf of all the authors. I will try to reply to all the points hoping to dispel any doubt or concern.

Starting from the major comment, I understand your point and I recognize that the wind tunnel signals filtering procedure and its consequences have not been clarified enough. As you have noticed, a relevant harmonic component around 4Hz is indeed present in the spectrum of the experimental thrust oscillation shown in Fig.7a (its PSD is in attachment as Fig.1). However, there are strong evidences suggesting such peak to be due to an interference with the electric motor of the nacelle. In fact, such harmonic is

present in all the tests and corresponds to the rotor's revolution frequency (1P), which is similar to the second surge harmonic only in case 59 (see also the PSD of test 53 in Fig.2). Its amplitude appears rather independent on the surge parameters. Furthermore, this component is only captured by the ATI balance on top of the nacelle, whilst it is almost absent in both the accelerometer signal (also on the nacelle) and the RUAG balance measurement (at the tower base). In the surge tests with steady rotor (NOW) this harmonic is not present, while a small peak at the tower's first bending mode is always observable around 6 Hz, although in the SIW case it is strongly smoothened by the aerodynamic damping. The ATI balance measurements for the cases 53 and 59 are Reported in Fig.3. It is evident that the peak at the surge frequency is dominant and the only reason why the others become so important in the aerodynamic thrust is that the inertia subtraction procedure only works at the surge harmonic. The other frequency components could even be amplified subtracting inertia with Eq. 2 (depending on the phases). Therefore, after the inertia subtraction the only meaningful signal becomes the one at the surge frequency.

Despite a 1P harmonic may also be triggered by aerodynamic effects, its insensitivity to the surge parameters together with the lack of a corresponding peak in the accelerometer measurement suggest that it might have been just a disturbance, likely from the electric motor. The absence of such harmonic in all the numerical results further confirms this hypothesis. In fact, the high fidelity CFD models are expected able at least to capture the unsteady aerodynamic effects, albeit with moderate accuracy. Finally, there is no trace of significant 1P harmonics in the literature data as well.

About the eventuality that the numerical models may not be able to capture phenomena observed in the experiments I deem it very unlikely. In our case the only relevant assumption common to all the codes is the modelling of the turbine as rigid. However, the modal analysis conducted on the scaled turbine suggests that if some unmodelled aeroelastic effect occurred, it would occur at higher frequencies than 1P.

Such an in-depth analysis of the wind tunnel tests does not fit the scope of the paper

under review. Nevertheless, some more comments will be added in hope to clarify the reasonable remarks that you made.

The choice to focus on the surge motion was made for the great advantages it gives in terms of kinematics. Being the first experimental campaign featuring such a level of detail, it was wisely decided to start from the easiest case, which has revealed not so easy anyway. Of course, the topic of pitch remains of paramount importance and it will be hopefully addressed in future work.

In the revised version I will try to put some more emphasis on the platform motions, a bit shortening the part on control.

About the names RATED1 and RATED2 I confirm what written in the previous reply.

Concerning the lifting line comment in line 175, this point can be subject to debate as all BEM codes condense the blade to a lifting line and treat the wake using momentum theory. As such, it makes use of lifting line variables as 2D airfoil coefficients and induction. To prevent confusion the sentence has been reformulated anyway.

Section 3.1 gives a high-level overview of the available models. The actual models and effects relevant for the simulations in the paper are discussed in section 3.1.1 to 3.1.3. Sentences in section 3.1.3 will be added to provide extra info. Text will be also added to clarify the choice of the simulation parameters that had been validated before through a sensitivity analysis indeed.

In Section 3.2 the LES simulations settings are described. Being a free flow without BL (the wind tunnel walls are modelled as smooth), the Popes' criterion was verified by empirically estimating the integral length of the free inflow turbulence and choosing the mesh size accordingly. A similar comment may be added to the revised version as well.

Additional information about the modeling of the surge motions in the full CFD simulations will be added to Section 3.3.

Table 4 and Figure 7 will be adjusted following your suggestions.

The factor $e^{(i2\pi f_s t)}$ has been implied in Eq. 6 and 19, but I agree that this should be specified.

I hope this answer will be enough to clarify your doubts, otherwise I remain available to provide you with additional data or comments. I will do my best to revise the use of English in the manuscript. Thank you very much for your review.

Yours faithfully,

Simone Mancini
* * *
**Thrust PSD**

**Fig. 1.**

**Thrust PSD**

[Figure]

**Fig. 2.**

[Figure]

[Figure]

**Fig. 3.**

---

## Editor Comment (EC1) · Alessandro Bianchini (Editor) · 28 Sep 2020

Dear authors, I have gone through your responses and I am asking you to submit a revised version of the manuscript at your earliest convenience. In doing so, I am also suggesting to pay attention to the style of your writing in order to improve the paper as a whole, since data are very interesting. In particular: 1) please try to make the style of the different sections more uniform. 2) I would suggest a revision from a native English speaker. Some sentences are not properly constructed and often there is a wrong use of articles and prepositions. I look forward to seeing your revised paper. Best regards,

---

## Author Response (AR1)

**Authors' response and changes to the manuscript**

- I would suggest precising the title to "Characterization of the unsteady aerodynamic response of a floating offshore wind turbine to surge motion".
The title has been changed similarly to what suggested.

- Maybe it should be clarified early in the paper that whenever surge is mentioned, harmonic surge is considered.
The adjective harmonic has been used more extensively when the imposed surge motion has been mentioned.

- Line 18: The variables in the equation should be introduced.
Corrected.

- Line 73: The bracket before "(Bayati et al., 2017b" should probably be moved "Bayati et al. (2017b".
Corrected.

- Figure 2: Size should be increase for better readability.
The figure (now number 3) has been enlarged up to text width.

- Line 160: "i.e. all what attached" -> "i.e. all what is attached".
Rephrased and corrected.

- Line 294: double "the".
Corrected.

- Figure 7: Which test number or amplitude is shown?
The test number is specified in both the text and the caption.

- Line 310: a comma behind "both" is missing.
Corrected.

- Line 347: This sentences is hard to understand and should be rephrased: "The main cause of these oscillations is the turbulence that, albeit weak because of the smooth flow boundary condition at inlet, forms upstream the turbine because of both the high wind tunnel Reynolds number and the influence of the actuator forces in the rotor plane".
The sentence has been rephrased.

- Line 449: "…coherent to what was observed for the thrust…".
Rephrased and corrected.

- Figure 7a: In the raw WT timeseries a frequency double the surge frequency can be clearly seen. Please explain where this frequency originates and why it was filtered out. This is quite important as the good agreement between numerical codes and experiments would not be achieved otherwise. Furthermore, this is crucial to justify the implicit assumption that thrust variations due to surge have the same frequency of the surge variation made in section 4.2.2. The filtering procedure should also be explained in further detail as the signal was filtered quite significantly.
Section 4.2.1 has been updated, reformulating the comments on the unfiltered time history for better clarity (the details have been provided in the discussion).

- Conclusions: "The codes have all confirmed the aerodynamic response to be dominated by the component at the surge frequency. Hence, considering only that harmonic, it has been possible to clean the experimental measurements that were characterized by significant disturbances due to the unsteady tests' complexity." – Could it be possible that the numerical models are not able to capture phenomena observed in the experiments? Please elaborate on this point.
This point has been covered in the Authors' comment posted in the discussion.

- English should be checked thoroughly. For instance, the preposition "a" and the article "the" are often missing or used inappropriately.
The use of English has been extensively revised with the aid of a native speaker.

- Introduction: The authors mention that the developed test rig has 2 degrees of freedom: pitch and surge. This study seems to be focused only on surge however, can you explain this choice in more detail? Furthermore, since the work focuses on basic understanding of the aerodynamic phenomena & code performance, it would be useful to the uninitiated if an explanation of the most aerodynamically relevant platform motions are and perhaps a diagram refencing those motions. At the very least authors should provide references for the mentioned information.

Some sentences have been added in the introduction, along with a schematic figure, hoping to give a clearer picture of the topic to the uninitiated.

- The introduction also mentions the lack of the influence of floating dynamics on WT control. In this paper however pitch control is disabled. I suggest shortening the paragraph and only mentioning that the results are presented in the frequency domain as well that is useful for controller design.

That part has been shortened as suggested.

- Section 2.2: I find the names RATED1 and RATED2 confusing. Although they both refer to design TSR conditions, one refers to below-rated wind speeds. Controller behavior can be very different at rated and below rated. In accordance with the paper, I suggest the names to be changed to BELOW and RATED. I am open to other reasonable explanations.

The names have been changed as suggested.

- Line 175: Please revise the phrase "Both models are lifting line codes, i.e. they make use of aerodynamic look-up tables to evaluate airfoil performance." The fact that a code uses aerodynamic look-up tables does not necessarily mean that it is a lifting-line code, actually BEM codes are typically not LL codes. If the specific code includes a lifting-line formulation for the blades and momentum modelling for the wake it should be clarified.

The sentence in question has been removed.

- Section 3.1 Please clarify the effects that are being modelled with engineering models in BEM. This is crucial for a fair comparison.

Sentences have been added in Section 3.1.3 to clarify this point.

- Section 3.1.3 Have the values discussed here been validated by means of a sensitivity analysis? Other authors have suggested much shorter timesteps and longer wakes to obtain independent results.

Sentences have been added in Section 3.1.3 to clarify this point as well.

-Section 3.2 The authors mention that a LES simulation was performed. Was the Pope criterion or similar criteria to verify that an adequate percentage of the turbulent spectrum was resolved verified?

As anticipated in the discussion, a sentence has been added to specify that the characteristic cell size was in the integral range of turbulence in the inflow.

- Section 3.3 How is the surge motion modelled in the CFD code? Please specify if automatic remeshing or grid deformation is applied or if there are rotating interfaces as sometimes seen when simulating rotors.

A sentence has been added to clarify this point.

- Figure 5: The authors might already be aware of this but it would be useful to include curves for the "full CFD" model a swell, to better highlight which model over-under estimates power & torque.

As explained in the discussion, the load distributions from the full CFD were not available unfortunately.

- Table 4: Please clarify the parameters fs and As in the description.

Corrected.

- Figure 7: QS timeseries is had to make out, please choose another color.

Several alternative colors have been tested, but none of them gave a satisfactory result and thus the color has been left yellow. However, a different dash style has been employed to improve readability.

- Section 4.2.2 It seems to me that in the formulas 6 and 7 the dependency from e^(2*pi*fs) was omitted. Please clarify this point. The same considerations apply to eq. 19.

The term has been implied in the phasor representation and this has been specified in the text.

[revised manuscript text omitted]